# Dual transcriptome based reconstruction of Salmonella-human integrated metabolic network to screen potential drug targets

**Kadir Kocabaş**[1], **Alina Arif**[2], **Reaz Uddin**[2], **Tunahan Çakır**[1]*

**1** Computational Systems Biology Group, Department of Bioengineering, Gebze Technical University, Gebze, Turkey, **2** Dr. Panjwani Center for Molecular Medicine and Drug Research, International Center for Chemical and Biological Sciences, University of Karachi, Karachi, Pakistan

* tcakir@gtu.edu.tr

**Data Availability Statement:** The codes are provided as a supplementary file. The transcriptome data was downloaded from Gene Expression Omnibus (GSE117236).

## Abstract

*Salmonella enterica* serovar Typhimurium (*S*. Typhimurium) is a highly adaptive pathogenic bacteria with a serious public health concern due to its increasing resistance to antibiotics. Therefore, identification of novel drug targets for *S*. Typhimurium is crucial. Here, we first created a pathogen-host integrated genome-scale metabolic network by combining the metabolic models of human and *S*. Typhimurium, which we further tailored to the pathogenic state by the integration of dual transcriptome data. The integrated metabolic model enabled simultaneous investigation of metabolic alterations in human cells and *S*. Typhimurium during infection. Then, we used the tailored pathogen-host integrated genome-scale metabolic network to predict essential genes in the pathogen, which are candidate novel drug targets to inhibit infection. Drug target prioritization procedure was applied to these targets, and pabB was chosen as a putative drug target. It has an essential role in 4-aminobenzoic acid (PABA) synthesis, which is an essential biomolecule for many pathogens. A structure based virtual screening was applied through docking simulations to predict candidate compounds that eliminate *S*. Typhimurium infection by inhibiting pabB. To our knowledge, this is the first comprehensive study for predicting drug targets and drug like molecules by using pathogen-host integrated genome-scale models, dual RNA-seq data and structure-based virtual screening protocols. This framework will be useful in proposing novel drug targets and drugs for antibiotic-resistant pathogens.

## Introduction

*S*. Typhimurium, is a gram-negative invasive and facultative pathogen that can infect various animal species [1]. Upon infection, it mostly causes food poisoning and leads to gastroenteritis in humans [2]. *Salmonella* infection causes 130,000 deaths every year, and it mostly affects people in low income countries [3]. *S*. Typhimurium is an intracellular pathogen, residing inside a membrane-bound compartment within host cells during infection [4]. This compartment is called Salmonella-containing vacuole (SCV), and it enables proliferation of *S*. Typhimurium inside the host cell by escaping from the host defense mechanism. Although SCV is considered a nutrition poor environment, this does not pose a problem for *S*. Typhimurium due to its

**Funding:** This work was supported by TUBITAK, The Scientific and Technological Research Council of Turkey (Project Code: 316S005) and by PSF, The Pakistan Science Foundation [Project Code: PSF-TUBITAK/S-HEJ (04)]. The funders had no role in study design, data collection and analysis, decision to publish, or preparation of the manuscript.

**Competing interests:** The authors have declared that no competing interests exist.

highly adaptive lifestyle [1]. Besides being a highly adaptive pathogen, the increasing rate of antibiotic resistance by *S*. Typhimurium has the potential to become a serious concern for public health. Therefore, it is important to identify novel drug targets to eliminate *S*. Typhimurium infections [5]. Understanding the metabolic activities of a pathogen is an important part of drug development process. Transcriptome data is a key data type that can elucidate metabolic activities of an organism [6]. It reflects enzymatic activities of the organism via mRNA levels that carry genetic information of those enzymes. One of the novel approaches in transcriptomics is dual RNA-sequencing (dual RNA-seq), which can be used to elucidate pathogen-host relationship since it measures mRNA levels of pathogen and host simultaneously during infection [7, 8].

Genome-scale metabolic network (GMN) models have shown utility in the analysis of metabolic activities of pathogen and host during infections [7, 9] Analysis of GMN models with constraint-based techniques to predict novel drug targets has the advantages of (i) being cost effective, (ii) being time efficient, (iii) providing a wide range of analyses of metabolic pathways at the same time. There are several studies about the prediction of novel drug targets to eliminate pathogen induced infections by analyzing GMN models [10]. Most widely used constraint-based computational approach for the analysis of GMNs is Flux Balance Analysis (FBA). FBA is a mathematical optimization technique that uses linear optimization to predict distribution of metabolic fluxes at steady state conditions. It uses an objective function besides constraints to select an optimum point from the flux solution space [11]. In silico gene deletion analysis is another widely used constraint-based analysis technique, which is used to determine potential drug targets [12–18]. Several GMN models were reconstructed so far for different *Salmonella* strains [19–21]. But these models are generic models, and they do not represent the metabolism of *Salmonella* inside a host cell. There are several techniques to create condition specific GMN models by mapping transcriptome data on to the generic GMNs. One of the commonly used transcriptome data mapping methods to generate condition specific GMNs is Gene Inactivity Moderated by Metabolism and Expression (GIMME) algorithm [22], which predicts active and inactive reactions in a GMN based on mRNA levels belonging to a particular condition.

Infection leads to a set of intricate interactions between pathogen and host cells, and these interactions should be taken into account in the process of identification of novel drug targets [9]. Pathogen-host integrated GMNs have potential to shed light on pathogen-host interactions (PHI) when integrated with dual RNA-seq data [7]. Pathogen-host metabolic modeling is a multi-cellular interaction modelling approach, where GMNs of both pathogen and host organisms are integrated in the simulations of metabolic phenotypes. Even if not commonly used yet, some pathogen-host GMNs are available in the literature [23, 24]. Here, we aim to provide a better insight into *S*. Typhimurium and host interactions by taking advantage of pathogen-host integrated GMNs and dual RNA-seq data in order to determine novel drug targets that can eliminate *S*. Typhimurium induced infections. We further report potential drugs for the identified drug target candidates by using a drug repositioning based approach. Through a prioritization criteria, pabB was selected as a high-ranked putative target, and a structure-based screening of novel drugs for pabB was performed using molecular docking simulations. To our knowledge, this is the first study that reconstructs a condition-specific pathogen-host GMN model by mapping dual RNA-seq data to predict novel drug targets.

## Results

### Pathogen-host integrated genome scale metabolic network analysis

A pathogen-host integrated GMN model was reconstructed in this study for the first time in literature for *S*. Typhimurium, with a total of 3586 genes from both organisms and 11,029

**Table 1. The reaction and metabolite numbers of condition specific pathogen-host GMN models.**

|  | Condition-specific GMN at the beginning of infection | Post-infection condition specific GMN at 8th hour | Post-infection condition specific GMN at 16th hour |
|---|---|---|---|
| Number of Reactions | 8773 | 8933 | 9089 |
| Number of Metabolites | 6941 | 6982 | 6979 |
| Number of HeLa reactions | 6595 | 6681 | 6536 |
| Number of *S.* Typhimurium Reactions | 2178 | 2252 | 2553 |

reactions. This model was used to generate condition-specific pathogen-host GMN models by mapping the dual RNA-seq data from $0^{th}$, $8^{th}$ and $16^{th}$ hours of infection [25]. GIMME was used to integrate the model and the transcriptome data, and the number of reactions for the corresponding condition-specific pathogen-host GMNs are given in Table 1.

Pathogen-host GMN was created by combining generic human and *S.* Typhimurium GMN models, and the number of reactions decreased in the condition-specific pathogen-host GMN models since they represent infected HeLa cell by *S.* Typhimurium in specific conditions. Nearly 2500 reactions were discarded from the generic Pathogen-Host GMN model since they were not active at the beginning of infection based on mRNA levels. On the other hand, the reaction profile of condition-specific pathogen-host GMN is different between the beginning and the late stages of infection. As infection progresses, the number of HeLa reactions remains almost the same while *S.* Typhimurium reactions dramatically increase (Table 1). The reaction profiles at the beginning of infection ($0^{th}$ hour) and at late stage of infection ($16^{th}$ hour) were compared. Even if there is not a considerable change in the number of HeLa reactions, the reaction profiles are different between the two conditions (Fig 1). During the infection, *S.* Typhimurium must adapt to the nutrition environment and physical conditions to survive inside the host [26]. Therefore, the increase in the number of *S.* Typhimurium reactions during the infection can be attributed to the activation of genes that might be necessary for the adaptation and proliferation of the pathogen. The reaction profiles of condition-specific pathogen-host GMNs were compared to each other in order to identify alterations in metabolic pathways based on the progress of infection. For host and pathogen separately, reactions only active in the condition-specific GMN at the beginning of infection ($GMN^{0th}$) and only active in the post-infection condition-specific GMN at the $16^{th}$ hour ($GMN^{16th}$) were identified and grouped by their pathways. The identified metabolic pathways and corresponding number of infection-time specific reactions are given in Fig 1.

Fig 1. shows that there is severe modulation in the lipid related pathways of host in infection. Lipid related pathways of the host are known to be subjected to modulation during the invasion of bacteria [27]. On the other hand, multiple metabolic pathways of *S.* Typhimurium are altered during infection based on Fig 1. Most dramatic change is in glycerophospholipid metabolism, which is one of the most important pathways for dual-membrane envelope of gram-negative bacteria [28]. There is also a dramatic change in alternate carbon metabolism, implying the utilization of different carbon sources other than glucose in the late stage of infection. Even if glucose is major carbon source for *S.* Typhimurium, it utilizes different carbon sources during infection [29]. Cofactor and prosthetic group biosynthesis, which is tightly related to enzymatic activities, is also altered as expected since enzymatic activities become varied as infection progress.

The first step in the validation of GMN models is comparing predicted flux rates with experimental data from literature. FBA was performed to predict flux rates of GMN models by maximizing TB (see Material and Methods). Ethanol and succinate production rates were set

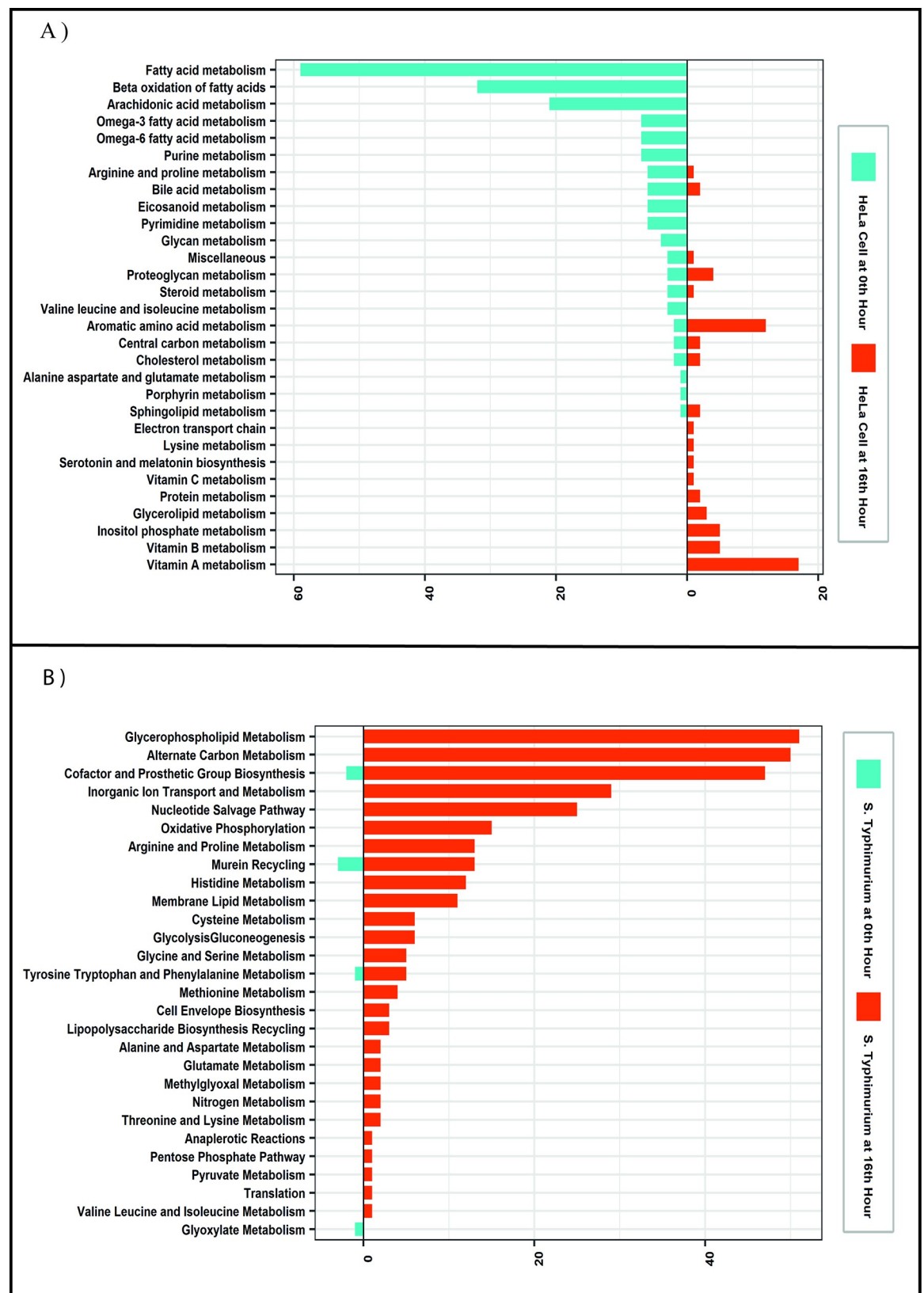

**Fig 1. The differences in reaction profiles of condition-specific pathogen-host GMNs grouped by pathways.** Pathways with at least 2 differential reactions are given. Red and green bars represent the number of HeLa cell reactions that are only active in 0[th] hour and 16[th]

hour respectively. Purple and blue bars represent the number of *S*. Typhimurium reactions that are only active in $0^{th}$ and $16^{th}$ hour respectively. The pathway names are listed on the left side of figure.

to zero based on literature information [30], and the other constraints were set as detailed in the section Materials and Methods. Predicted secretion rates of major by-products for *S. typhimurium* are given in Table 2 together with the literature-reported secretion rates at infection. The relative rates of by-products predicted by the pathogen-host integrated condition-specific model at $16^{th}$ hour are in perfect agreement with the experimental data from HeLa-infecting *S.* Typhimurium cells [30]. Repeating simulations by using the non-reduced model leads to rates about 20% higher than the reduced model predictions. The use of only reduced *Salmonella* model without the host network, on the other hand, led to 25% higher acetate secretion rates. Predicted acetate, formate, and lactate secretion rates at $8^{th}$ hour of infection, on the other hand, are very close to the rates predicted for the beginning of infection (Fig 1). Therefore, for the rest of the study, we used the model reconstructed for the $16^{th}$ hour of infection as the model representative of the infectious state of the organism.

### Identification of potential drug targets

Prediction of essential genes (EG) for the survival of pathogen inside host organism is the primary step for most of the drug discovery processes. Enzymes produced from EGs are potential drug targets that can be targeted with chemical molecules to eliminate the pathogen. EGs for the infection were predicted by using GMN$^{16th}$, which represents the infectious state. Here, 140 EGs were predicted for the infection. Predicted 140 EGs were compared with literature by using Database of Essential Genes (DEG) [31], which reports data from three experimental gene deletion studies from rich medium experiments [32–34] Data were available for 137 of 140 predicted EGs, 93 of which were reported as essential genes in at least one study (68%). (S1 Table). The genes falsely predicted as essential can be attributed to the fact that the experiments were performed in rich medium conditions with no host cells involved while the simulations were performed by the pathogen-host integrated genome-scale metabolic network.

Drug targets should not show high amino acid sequence similarity with human proteins in order to prevent side effects. Therefore, homology analysis was performed to identify drug targets that are similar to human proteins. The similarity was determined based on the predefined cutoff value detailed in Materials and Methods. Out of 140 potential drug targets, 52 proteins were discarded since they have high similarity with human proteins (S2 Table). Pathway enrichment analysis was performed with non-homologous 89 proteins in order to characterize

**Table 2. Predicted flux rates obtained by FBA analysis of condition-specific pathogen-host integrated GMNs are compared with the experimental results.**

|  | Infection ($0^{th}$ hr) Flux Values (mmol/gDW/h) | Infection ($8^{th}$ hour) Flux Values (mmol/gDW/h) | Infection ($16^{th}$ hour) Flux Values (mmol/gDW/h) | Experimental (nM/cell/h) [30] |
|---|---|---|---|---|
| **D-Lactate Secretion** | 0 | 0 | 11.29 | 10 ± 3 |
| **Acetate Secretion** | 1.94 | 2.17 | 6.80 | 4 ± 2 |
| **Formate Secretion** | 10.86 | 9.14 | 3.55 | 2 ± 1 |
| **Succinate Secretion** | 0 | 0 | 0 | 0 |
| **Ethanol Secretion** | 0 | 0 | 0 | 0 |

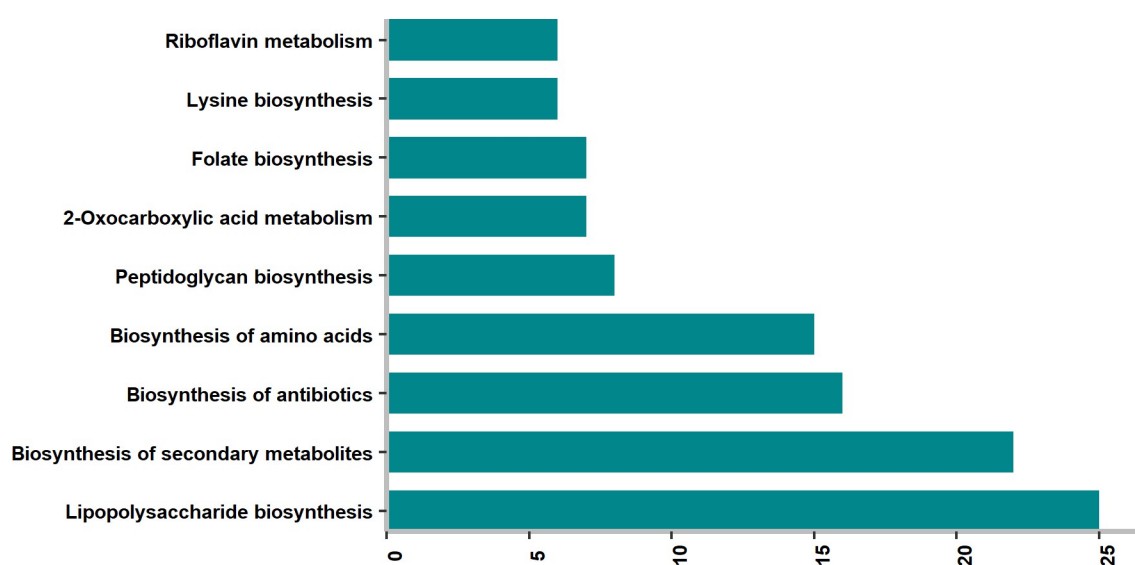

**Fig 2. Pathway enrichment analysis result.** The values on the x axes indicates number of drug targets in the related pathway with no homology to human proteins.

potential drug targets (Fig 2). The most enriched pathway is the lipopolysaccharide biosynthesis pathway, which is critical for the survival of *S*. Typhimurium since it maintains the functionality of the outer membrane of the pathogen [35]. Another enriched pathway is the biosynthesis of amino acids, and it is a reasonable metabolic pathway that can be targeted since it is indispensable for pathogens [36]. Like lipopolysaccharides, peptidoglycans are also indispensable molecules for the functionality of bacterial cell wall [37], and their biosynthesis pathway was also captured (Fig 2). Consequently, the general composition of enriched pathways indicates that targeted proteins serve in the pathways that are crucial for amino acid and cell membrane metabolisms.

Druggability analysis was performed to identify potential drug targets that can be targeted with chemical molecules. The druggability analysis aims to identify proteins that have a high affinity to bind to drug-like molecules since some proteins do not have this property [18]. Here, 43 potential drug targets that have high affinity to bind drug-like molecules were determined among 89 non-homologous pathogen proteins (S3 Table). Determining potential drug targets that are broadly distributed among other harmful bacteria is one of the important steps of the prioritization process. [18]. To identify such potential drug targets, broad-spectrum analysis was performed. Finally, 28 potential drug targets that are non-homologous to human proteins, druggable and broadly distributed among other bacteria were determined. The list of final potential drug targets is given in Table 3 and S4 Table. Of 28 predicted potential drug targets, 20 were reported as essential in the DEG database. When we ranked the drug target list in terms of broad spectrum score, i.e. number of pathogenic bacteria with significantly similar sequence of the gene, 16 of the top 20 genes were essential based on DEG (80%).

## Analysis of the prioritized drug targets

The prioritized drug targets (Table 3) were clustered into pathways that are crucial for the survival of the pathogen. There are four proteins, pabB, folA, folK, folP, that take part in folate biosynthesis pathways. For nearly all organisms, the folate biosynthesis pathway is one of the indispensable pathways in order to maintain life. Folates are necessary for the production of

**Table 3. Model-derived potential drug targets that obey three criteria: No homology to human proteins, druggable, and broad-spectrum behaviour.**

| Locus Names | Gene Symbol | Protein Name | Pathway | Reported at DEG |
|---|---|---|---|---|
| STM1824 | pabB | Aminodeoxychorismate synthase component 1 | Folate biosynthesis | Yes |
| STM0087 | folA | Dihydrofolate reductase | Folate biosynthesis | Yes |
| STM0183 | folK | 2-amino-4-hydroxy-6-hydroxymethyldihydropteridine pyrophosphokinase | Folate biosynthesis | No |
| STM3295 | folP | Dihydropteroate synthase | Folate biosynthesis | No |
| STM0064 | dapB | 4-hydroxy-tetrahydrodipicolinate reductase | Biosynthesis of amino acids, L-lysine biosynthesis via DAP pathway | Yes |
| STM0213 | dapD | 2,3,4,5-tetrahydropyridine-2,6-dicarboxylate N-succinyltransferase | Biosynthesis of amino acids, L-lysine biosynthesis via DAP pathway | Yes |
| STM0207 | mtnN | 5'-methylthioadenosine/S-adenosylhomocysteine nucleosidase | Biosynthesis of amino acids, Cysteine and methionine metabolism. | No |
| STM3486 | aroB | 3-dehydroquinate synthase | Biosynthesis of amino acids, Phenylalanine, tyrosine and tryptophan biosynthesis | No |
| STM2384 | aroC | Chorismate synthase | Biosynthesis of amino acids, Phenylalanine, tyrosine and tryptophan biosynthesis | No |
| STM3862 | glmU | Bifunctional protein GlmU | UDP-N-acetyl-alpha-D-glucosamine biosynthesis | Yes |
| STM2094 | rmlC | dTDP-4-dehydrorhamnose 3,5-epimerase | Polyketide sugar unit biosynthesis, Streptomycin biosynthesis | No |
| STM1772 | kdsA | 2-dehydro-3-deoxyphosphooctonate aldolase | Lipopolysaccharide biosynthesis | Yes |
| STM3316 | kdsC | 3-deoxy-D-manno-octulosonate 8-phosphate phosphatase KdsC | Lipopolysaccharide biosynthesis | No |
| STM0988 | kdsB | 3-deoxy-manno-octulosonate cytidylyltransferase | Lipopolysaccharide biosynthesis | Yes |
| STM0310 | gmhA | Phosphoheptose isomerase | Lipopolysaccharide biosynthesis | No |
| STM0228 | lpxA | Acyl-[acyl-carrier-protein]—UDP-N-acetylglucosamine O-acyltransferase | Lipopolysaccharide biosynthesis | Yes |
| STM0134 | LpxC | UDP-3-O-acyl-N-acetylglucosamine deacetylase | Lipopolysaccharide biosynthesis | Yes |
| STM1200 | tmk | Thymidylate kinase | Pyrimidine metabolism | Yes |
| STM1707 | pyrF | Orotidine 5'-phosphate decarboxylase | Pyrimidine metabolism | Yes |
| STM1426 | ribE | Riboflavin synthase, alpha chain | Riboflavin metabolism | Yes |
| STM0417 | ribH | 6,7-dimethyl-8-ribityllumazine synthase | Riboflavin metabolism | Yes |
| STM0045 | ribF | Riboflavin biosynthesis protein | Riboflavin metabolism | Yes |
| STM3307 | murA | UDP-N-acetylglucosamine 1-carboxyvinyl transferase | Peptidoglycan biosynthesis. Amino sugar and nucleotide sugar metabolism | Yes |
| STM0129 | murC | UDP-N-acetylmuramate—L-alanine ligase | Peptidoglycan biosynthesis. D-Glutamine and D-glutamate metabolism | Yes |
| STM0123 | murE | UDP-N-acetylmuramoyl-L-alanyl-D-glutamate—2,6-diaminopimelate ligase | Peptidoglycan biosynthesis.Lysine biosynthesis | Yes |
| STM0128 | murG | UDP-N-acetylglucosamine—N-acetylmuramyl-(pentapeptide) pyrophosphoryl-undecaprenol N-acetylglucosamine transferase | Peptidoglycan biosynthesis | Yes |
| STM0124 | murF | UDP-N-acetylmuramoyl-tripeptide—D-alanyl-D-alanine ligase | Peptidoglycan biosynthesis.Lysine biosynthesis | Yes |
| STM3725 | coaD | Phosphopantetheine adenylyltransferase | Pantothenate and CoA biosynthesis | Yes |

essential biomolecules such as nucleic acids and amino acids. Most bacteria, fungi and plants can synthesize folate, while animal cells take it up from external sources [38]. This pathway is a potentially promising drug target since human cells do not have a folate synthesis mechanism that might be manipulated by a pathogen. Five of the prioritized drug targets have functionality in the biosynthesis of amino acids (dapB, dapD, mtnN, aroB, aroC). dapB and dapD take place in L-lysine biosynthesis via diaminopimelic acid (DAP) pathway, and the side product of this pathway is m-DAP, which is an essential biomolecule for peptidoglycan cell wall for gram-negative bacteria [39]. glmU has a role in the production of UDP-N-acetyl-alpha-D-glucosamine, which is essential for bacterial cell wall [40]. rmlC has an important role in the synthesis

of L-rhamnose, which is an important saccharide for the virulence of some pathogens including *S*. Typhimurium. The absence of L-rhamnose biosynthesis pathway in human cells makes this drug target more appealing [41]. The survival of bacterium depends on the integrity of cell envelope. kdsA, kdsC, kdsB, gmhA, lpxA and LpxC have roles in the production of lipopolysaccharides, which is critical for the formation of cell envelope [42]. tmk and pyrF are involved in pyrimidine metabolism, which is crucial for all living organisms. tmk catalyzes the phosphorylation of thymidine 5'-monophosphate, which is an essential reaction for pyrimidine synthesis [43]. ribE, ribF and ribH are required for the production of riboflavin, which is a precursor of flavin mononucleotide (FMN) and flavin adenin dinucleotide (FAD). Riboflavin synthesis is a pathway required for the survival of gram-negative bacteria in the absence of external riboflavin synthesis [18]. murA, murC, murE, murG and murF are involved in the synthesis of peptidoglycans, which is an essential ingredient for bacterial cell wall biogenesis [44]. murA catalyzes the first step of peptidoglycan biosynthesis, and deletion of murA leads to death of *Escherichia coli* and *Streptococcus pneumoniae*. Fosfomycin is an antibiotic that targets murA in order to kill bacteria [45]. Mur ligases are known to be attractive drug targets because of their role in bacterial cell wall formation [46]. Pantothenate is a main precursor of coenzyme A, and its absence leads to deficiency in bacterial growth. coaD is involved in the fourth step in the coenzyme A biosynthesis pathway, which was investigated before as a suitable antibiotic target [47].

## Identification of potential drugs for pabB

4-aminobenzoic acid (PABA) synthesis is an attractive antibiotic target since it is an essential biomolecule for many pathogens and it does not have a human counterpart. PABA has two main functionalities in the bacteria; (i) it is a substrate for folic acid pathway, which is critical for survival of pathogen, (ii) it is a precursor in coenzyme Q biosynthesis, which is essential for virulence [48, 49]. PABA is synthesized in two steps, and the first step is catalyzed by pabA and pabB enzymes by converting chorismate to 4-amino-4-deoxychorismate [50]. And, the second step is the production of PABA from 4-amino-4-deoxychorismate. pabB was detected as one of the putative drug targets in this study by our drug target prioritization pipeline. We specifically focused on pabB in the rest of our study as a drug target to eliminate *Salmonella* infections since it has critical functionality in PABA synthesis pathway, which is not represented in human cells. pabB was reported to be essential experimentally in the DEG database, and, among the identified drug targets with experimental validation (Table 3), it ranks 7th in terms of the number of pathogenic bacteria strains that carry a gene with high sequence similarity based on our broad-spectrum analysis. Additionally, we investigated the importance of pabB in the pathogen-host integrated GMN model. Interactions of the metabolites of the pabB reaction (chorismate, 4-amino-4-deoxychorismate, L-glutamate, L-glutamine) with the metabolites of other reactions were visualized by creating a metabolite-metabolite interaction network (S1 Fig). L-glutamate is directly related to numerous amino acids such as alanine, leucine and asparagine. On the other hand, 4-amino-4-deoxychorismate is indirectly related with the production of thymidine through tetrahydrofolate. It is also linked with the production of adenine through R-Pantoate. Therefore, DNA synthesis is dependent on the production of 4-amino-4-deoxychorismate through adenine and thymine synthesis which cannot be synthesized when pabB is inhibited.

There is a very similar protein to this putative target in *Escherichia coli*, which is also called pabB. Formic acid, which is widely used as an antibacterial agent in fodders, was reported in DrugBank as a compound that targets pabB in *Escherichia coli* (strain K12) [51]. To our knowledge, pabB was not offered or examined as a drug target for *Salmonella* species before. Hereby,

protein docking and molecular dynamic analysis were performed to determine novel molecules that can inhibit *S.* Typhimurium growth by binding pabB. The protein Aminodeoxychorismate synthase component 1, which belonged to *S.* Typhimurium (strain LT2 / SGSC1412 / ATCC 700720) with a UniProt ID: P12680 (PABB_SALTY), was taken for elaborative structural and functional studies. The 3D structure of the protein is the core requirement to carry out protein molecular docking studies and its unavailability necessitated the modeling of the three-dimensional structure. Herein, first step was the comparative 3D modeling of the query protein with the application of MODELLER Software. The crystal structure of 4-amino4-deoxychorismate (ADC) synthase (PDB ID: 1K0E) was chosen as a template as its percent identity was 76.38% and query coverage was 99% against P12680. The MODELLER yielded five 3D structures of P12680 protein, out of which the best model with higher accuracy was selected (S2 Fig) after thorough quality assessment using PROCHECK server. The Verify3D passed the modelled structure with 87% indicating 80% amino acids of the built model having score > = 0.2 in the 3D/1D profile. Further, the Ramachandran plot (S3 Fig) showed 92.7% residues in the most favoured region whereas only 0.3% residues were reported in disallowed region. Once the model was finalized, $Mg^{2+}$ ion was incorporated into the built 3D structure as magnesium ion is reported as the cofactor within the target protein in the UniProt indicating its role in the catalysis activity. Therefore, $Mg^{2+}$ ion was complexed near the active site residues reported in UniProt and in literature [48].

The DoGSiteScorer produced 10 probable binding pockets, out of which pocket number one was selected as the binding site to utilize for molecular docking. The selected pocket has a druggability score of 0.83, suggesting the prime region for drug binding. The binding pocket was analyzed in Chimera software [52], and validated as it covers the active site amino acid residues reported in the UniProt along with the amino acid residues coordinating with the $Mg^{2+}$ ion. The predicted binding site can be seen in (S4 Fig).

Once all the 54,000 drug-like compounds were screened against the target protein (P12680), the lowest binding energy conformation of each single 54,000 compound(s) was obtained. The overall binding energies ranked against the P12680 are represented through a histogram in Fig 3.

The thorough analysis of the results suggested that 1659 compounds showed promising binding free energy ranging from -12.42 kcal/mol to -9.04 kcal/mol, while 22,231 compounds having binding free energies within the range of -5.66 kcal/mol to -2.28 kcal/mol. Moreover, top ten compounds that were docked near the active site of the target protein were (having binding free energy of -12.42, -12.02, -11.92, -11.90, -11.91, -11.98, -11.77, -11.73, -11.42, -11.72) retrieved to further investigate their chemical interactions with amino residues of target protein. The aforementioned top ten compounds with their ZINC IDs are reported in (S5 Table).

The protein-ligand complex of each top ten compound was explored to inspect the chemical bonds and interactions occurring within the protein-ligand complex. LigPlot+ generates the atomic interactions taking place among the target protein and ligand (drug) through hydrogen bonds and hydrophobic contacts. Each of the ten protein-ligand complex analyzed in LigPlot+ has shown ligand interactions with the $Mg^{2+}$. The reported active site residues from UniProt (i.e. Lys275 and Glu259) were found to be interacting with the top ten ligands, out of which Lys275 can be seen making hydrogen bond with five ligands and hydrophobic contacts with two ligands. (Table 4). On the other hand, Glu259 makes hydrophobic contacts with six of the ligands. The most common amino acid residues interacting with top ten ligands through hydrogen bonding were identified as Thr277, Gly427, Glu440, Lys444, Lys275, Arg411 and Glu259. Finally, each of the amino acid residues making hydrogen bonds and hydrophobic contacts interaction with the top ranked 10 compounds is presented in Table 4

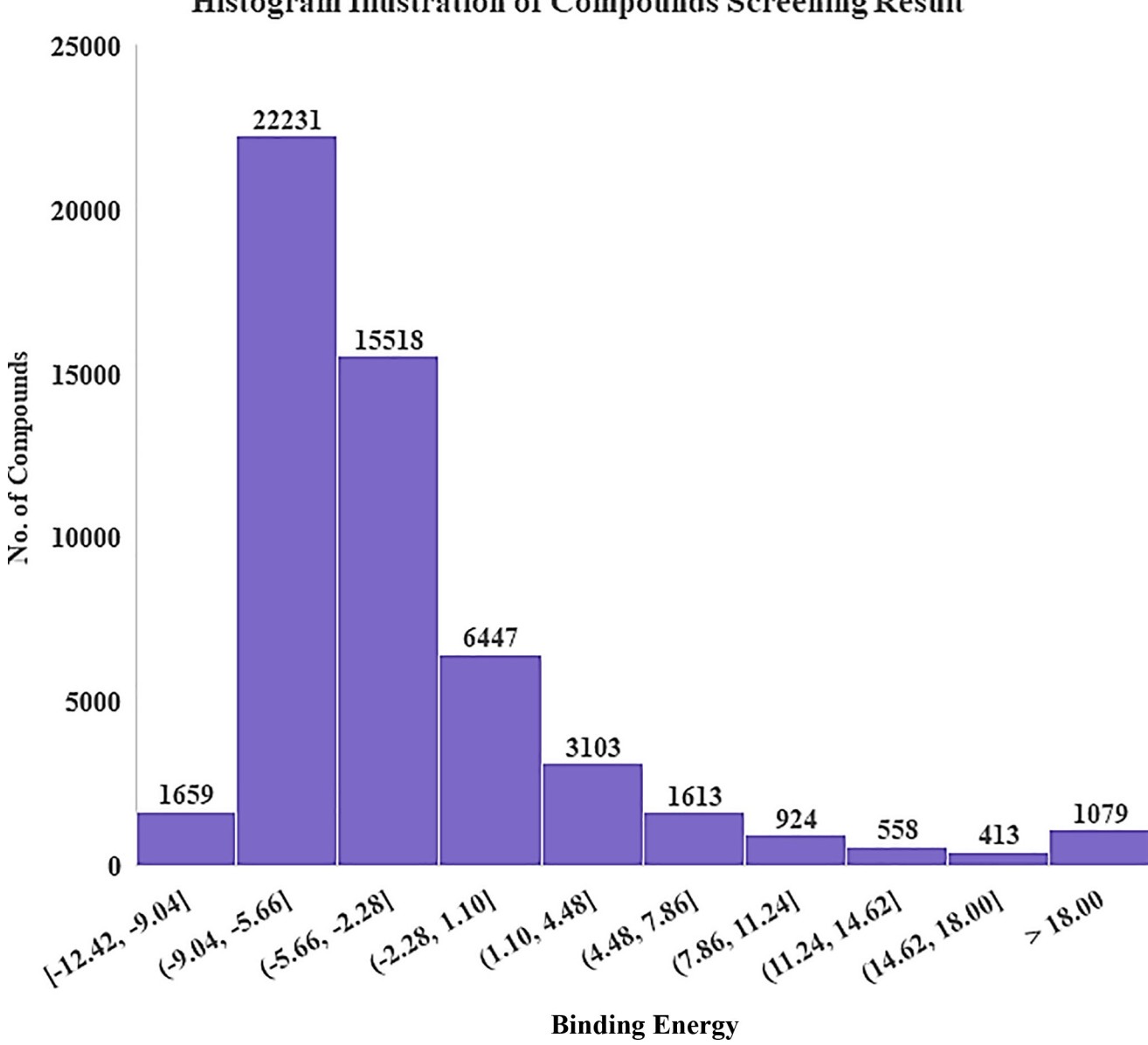

**Fig 3. Histogram illustration of the overall binding energy retrieved through virtual screening.**

and visualized in S5–S7 Figs. There is not any report in literature about the use of these compounds against pathogenic bacteria. Therefore, they remain open to experimental validation.

## Conclusions

Analysis of pathogen-host integrated GMN models is a well-suited approach in terms of identifying novel drug targets by considering pathogen-host interactions. It allows tracking the response of pathogen and host simultaneously during infection by mapping infection-induced dual-transcriptome data. There are some pathogen-host GMN models published [23, 24], but to our knowledge, this is the first study in the literature that determines drug targets by analysing condition-specific pathogen-host integrated GMNs created by mapping dual-

**Table 4. Protein residues involved in the hydrogen and hydrophobic interactions with top ten best ranking compounds (ligands), analyzed through LigPlot+.**

| S. No | ZINC IDs | Residues making hydrogen bond interaction | Residues making Hydrophobic contacts |
|---|---|---|---|
| 1 | ZINC7879733 | Thr277(A), Gly427(A) Lys444(A), Arg411(A) Lys275(A) | Ile410(A), Ala424(A), Gly427(A) Asn214(A) Trp391(A), Ser423(A), Gly426(A), Glu259(A) Ile368(A), Cys422(A), Gly425(A), Gly276(A), Ile274(A) |
| 2 | ZINC15179659 | Gly427(A), Arg411(A), Lys444(A), Thr277(A), Glu440(A) | Lys275(A), Trp391(A), Val445(A), OThr277(A) Ala424(A), Ser367(A), Ile410(A), Gly425(A) Ser423(A), Ile368(A), Ile448(A), Asn214(A), Gly426(A) Thr412(A), Cys422(A), GIle274(A), ly276(A) |
| 3 | ZINC14880941 | Thr277(A), Ser423(A) Gly427(A), Gly276(A) Glu440(A), Trp391(A) Asn214(A) | Lys444(A), Arg411(A), Val445(A), Ile410(A) Lys275(A), Ile274(A), Gly425(A), Ala424(A), Ile368(A) Gly426(A), Ile448(A), Glu259(A), Cys422(A) |
| 4 | ZINC58542694 | Arg411(A), Thr277(A), Glu440(A), Lys275(A), Gly427(A), | Lys444(A), Gly276(A), Trp391(A), Ile410(A), Ser423(A), Ala424(A), Ile274(A), Asn214(A), Ile368(A), Val445(A), Gly425(A), Gly426(A), His340(A), Ser367(A), |
| 5 | ZINC1201089024 | Trp391(A), Lys275(A), Asn214(A), Arg411(A), Glu440(A), Thr277(A), Gly427(A), | Lys444(A), Gly276(A), Ile410(A), Ile274(A), Gly425(A), Ile368(A), Val445(A), Glu259(A), Ser423(A), Gly426(A) Cys422(A), Ile448(A), |
| 6 | ZINC27071723 | Lys444(A), Trp391(A), Lys275(A), Glu440(A), Thr277(A), Gly427(A), | Arg411(A), Gly276(A), Ile274(A), Val445(A), Ser423(A), Gly425(A), Ile410(A), Cys422(A), Gly426(A), Thr412(A), Ala424(A), Ile448(A), Asn214(A), Glu259(A), Ile368(A), |
| 7 | ZINC7133393 | Arg411(A), Gly425(A), Glu440(A), Thr277(A), Gly427(A) | Lys444(A), Gly276(A), Trp391(A), Ile410(A), Ala424(A), Val445(A), Ser423(A), Glu259(A), Asn214(A), Gly426(A)Ser367(A) |
| 8 | ZINC7879735 | Gly427(A), Lys444(A), Arg411(A), Glu440(A), Thr277(A), | Lys275(A), Gly425(A), Gly276(A), Asn214(A), Ile274(A), Ala424(A), Ile410(A), Ser367(A), Trp391(A), Ser423(A), Gly426(A), Cys422(A), Val445(A), |
| 9 | ZINC58542238 | Gly427(A), Gly425(A), Glu440(A), Lys275(A), Thr277(A) | Arg411(A), Lys444(A), Gly276(A), Trp391(A), Ser423(A), Val445(A), Ile274(A), Ile410(A), Asn214(A), Ala424(A), Gly426(A), His340(A), Ile368(A), Thr412(A), Cys422(A), Ile448(A), |
| 10 | ZINC7538530 | Arg411(A), Glu440(A), Thr277(A), Gly427(A), | Lys444(A), Gly276(A), Lys275(A), Ile274(A), Gly425(A), Trp391(A), Ile410(A), Ser423(A), Ala424(A), Gly426(A), Asn214(A), Glu259(A), Cys422(A), Val445(A) |

transcriptome data. We here reconstructed and analyzed condition-specific integrated GMN models to identify novel drug targets for *S.* Typhimurium induced infections. We used prioritization steps to identify best suitable novel drug targets. After prioritization processes, we identified 28 putative drug targets, and pabB was chosen as a high ranked drug target based on our prioritization pipeline, literature information and novelty. Subsequently, homology and molecular docking analyses were performed to identify candidate compounds that inhibit pabB. The top ten compounds in terms of binding free energy were identified and reported. Consequently, analysing *S. enterica* metabolism inside the host cell has enabled us to comprehend metabolic alterations in both HeLa cells and *S. enterica* along with determining novel drug targets. Future studies may provide more arguments for the proposed drug targets and inhibitors in this study. This study can be used as a guideline for creating and analysing condition specific pathogen-host GMN models.

## Materials and methods

The flowchart of the pipeline followed in this study is given in Fig 4. Each step is detailed in the sections below.

### Transcriptome data

The dual RNA-seq data of infected HeLa cells and *S.* Typhimurium strain SL1344 [25] was downloaded from NCBI Gene Expression Omnibus (GEO) Database [53]. The dataset ID in the GEO database is GSE117236. The samples collected at the beginning of infection and the post-infection data at 8[th] and 16[th] hours were used in this study. Each time point included duplicate samples. Principal component analysis (PCA) was used to identify any possible outliers in the data, and no outliers were detected (S8 Fig).

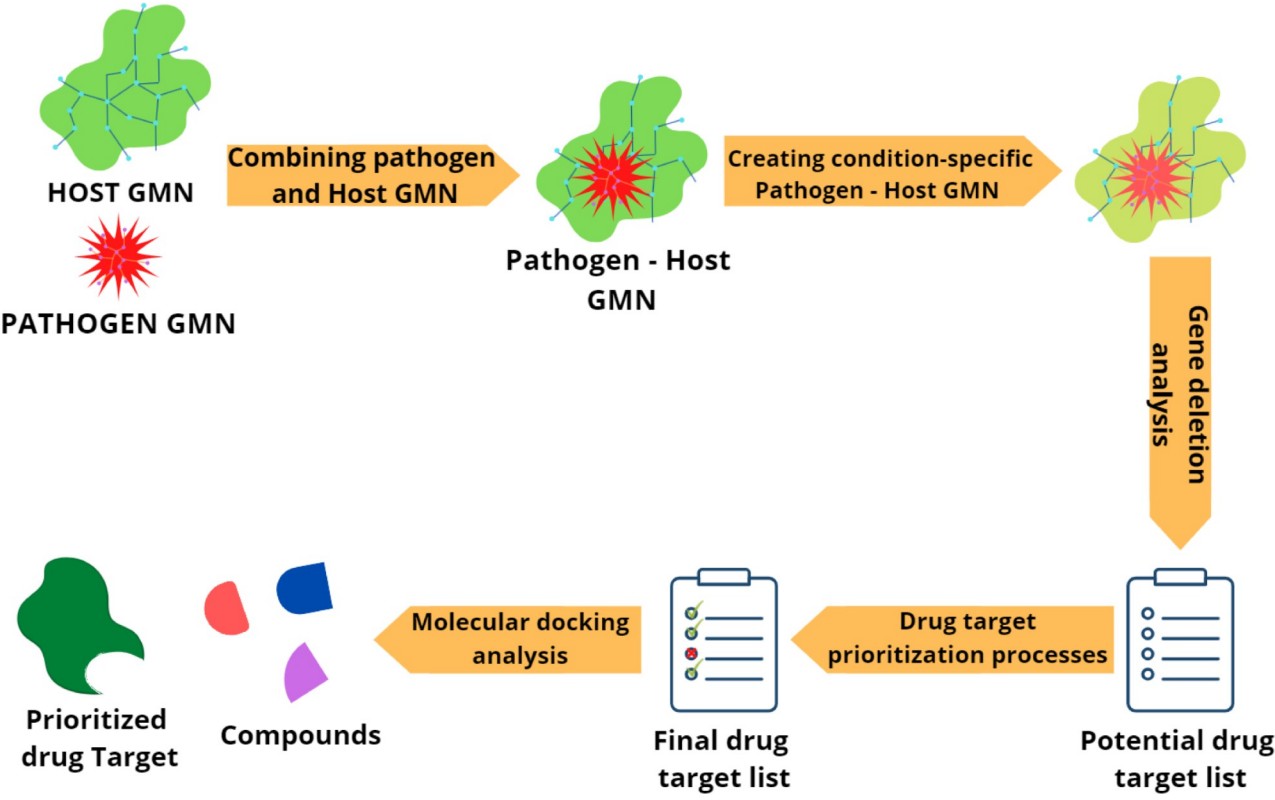

**Fig 4. The flowchart of the pipeline followed in this study.**

### Pathogen-host integrated genome scale metabolic network

Two different genome-scale metabolic models from the literature were used to reconstruct an integrated pathogen-host genome scale metabolic network (GMN). A genome-scale metabolic network of *S*. Typhimurium, called stm_v1.0, consisting of 2,545 reactions controlled by 1,271 genes was used as the pathogen metabolic network [21]. As the host GMN, a recent reconstruction of human metabolism with a substantial amount of curation, called iHsa, was used, which covered 8,336 reactions and 2,315 genes [54]. *S*. Typhimurium is an intracellular pathogen. Therefore, the GMN of the pathogen was placed inside the cytoplasm of the host GMN as a separate compartment to create the pathogen-host integrated GMN. Here, an extensive literature research was performed to identify cytosolic host metabolites that can be consumed by the pathogen during infection. An exchange reaction with the extracellular environment must be available for these metabolites in the *S*. Typhimurium metabolic network. 38 such metabolites were identified [26, 29, 55–57], and they were allowed to be taken up by the *S*. Typhimurium GMN from the cytoplasm of the host (S6 Table). In addition, the pathogen was allowed to secrete all its exchange metabolites to the host cytoplasm as defined by the model secretion reactions (S7 Table). As a result, the pathogen-host integrated GMN consisting of 11,029 reactions controlled by 3,586 genes was created by using COBRA Toolbox v.3.0 on MATLAB programming platform [58].

### Flux balance analysis

Pathogen-host integrated GMN was analyzed using flux balance analysis (FBA) method to predict fluxes associated with the infection times studied. FBA searches the solution space defined

by mass balance and reaction reversibility constraints to find an optimal solution with the help of an objective function. FBA assumes that the system is at steady state, i.e. the concentrations of intracellular metabolites do not change over sufficiently long time, leading to linear mass balance constraints [59]. The objective function of pathogen-host integrated GMN was defined based on a weighted relationship between host and pathogen biomass, and the biomass composition formulas were taken from iHsa and stm_v1.0 models (Eq 1) [21, 54, 60]. HB, PB and TB are host biomass, pathogen biomass and total biomass respectively in Eq (1), where α and β are maximum host and pathogen biomass production rates in order. Using the constructed pathogen-host GMN, α and β were calculated first with FBA via maximization of HB and PB reactions separately. Later, maximum host and pathogen biomass production rates (α and β), which were calculated as 0.39 and 0.28 respectively, were added to the equation as weights of HB and PB to get TB, which represents the balanced effect of HB and PB. Eq 1 was added as a reaction to the pathogen-host integrated GMN and later set as the objective function.

$$\propto \times HB + \beta \times PB = TB \qquad\qquad (Eq1)$$

The upper bound of glucose uptake rate of *S*. Typhimurium was set to 5 mmol/gDW/h in simulations based on the studies of Thiele and her coworkers [21]. The maximum uptake rate of other available carbon sources for *S*. Typhimurium inside the host cell was limited to 20% of its glucose uptake rate. Oxygen uptake rate of *S*. Typhimurium was set to 1 mmol/gDW/h to mimic the hypoxic environment during infection [61]. The host cell was allowed to utilize only metabolites that were found in the Dulbecco's Modified Eagle's Medium (DMEM) since the infection experiment for the dual RNA-seq data, which was used in creating condition-specific integrated GMN models, was carried out in this medium (S8 Table). In GMN models, alternate optima can be an issue in interpreting the results of FBA since there might be multiple flux distributions that result in the same value for the objective function. Minimization of the sum of squares of all flux values was applied to prevent alternate optima [62]. The principle of this method was proposed based on the accomplishment of the cellular goals with minimal resource expenditure since the flux values are an indication of the amount of depletion of resources [63].

## Integrating transcriptome data with pathogen-host integrated GMN

Condition specific GMNs were generated by mapping dual RNA-seq data on the pathogen-host GMN to simulate infection states at different time points. Gene Inactivity Moderated by Metabolism and Expression (GIMME) algorithm was used as the mapping algorithm to generate condition specific GMNs [22]. GIMME determines active reactions based on a threshold put on the mRNA levels in the data, where reactions below the thresholds are set as inactive. GIMME generates a GMN with the desired functionality using the objective fraction parameter, and it adds the reactions in the inactive set back if their removal affects the desired functionality [22]. The threshold value was determined as the quarter of the mean of the transcriptome data. Since the average gene expression values were much higher in HeLa cell compared to *S*. Typhimurium in the utilized dual RNA-seq data (S9 Fig), the threshold value was separately determined for both organisms. Then, since GIMME algorithm accepts a single threshold, the difference between the organism-specific threshold values were added to the reaction scores of *S*. Typhimurium, and the threshold value obtained from the human transcriptome was used as the threshold value in GIMME simulations. The objective fraction parameter was set to 0.2 in order to ensure that the condition specific integrated GMN produces at least 20% of the maximum TB. Three different condition specific GMNs were produced as a result to represent the start of infection (0th hour) and infections at 8th and 16th hours.

## Identification of drug targets

Gene deletion analysis is a widely used approach in constraint-based metabolic modeling to predict potential drug targets, and it is performed by in silico deletion of genes in the GMN [10]. The analysis aims to obtain essential genes for the desired functionality of a GMN, such as preventing growth of pathogen. FBA can be used to predict essential genes in an organism by setting the rate of the associated reaction(s) to zero for each gene. If inactivation of the reaction(s) lead to zero growth rate, the gene is essential for the pathogenic organism and it can be used as a drug target. In this study, gene deletion analysis was performed to identify potential drug targets that can restore the metabolic changes in the host cell caused by *S*. Typhimurium induced infection. GIMME-based condition-specific GMNs were used in the analysis. A non-zero rate for the production of HB together with zero rate for PB was used as the desired functionality of the condition specific GMNs in gene deletion analysis. Using the FBA technique to maximize HB and PB separately, essential genes were obtained, and the enzymes that catalyze these reactions were chosen as potential drug targets.

The identified drug targets were further filtered based on the approach applied elsewhere [18]. Briefly, the steps in the approach followed are as follows: (i) Homology analysis is a critical part of drug target selection process, and the aim of the analysis is determining drug targets that are not similar to host proteins in order to avoid side effects of drugs. Homology analysis was performed using BLASTp algorithm, and drug targets that are not similar to human proteins were identified [64]. As BLASTp parameters, cut off for the expected value (E-value) was chosen as $1x10^{-4}$, and the maximum sequence identity for the determination of human-non-homolog drug targets were chosen as 30% [18]. Pathway enrichment analysis was performed with the identified human-non-homolog drug targets by using KOBAS V3.0 [65] to elucidate pathways that are expected to be crucial for the survival of *S*. Typhimurium inside the host. (ii) Another important step in the model-based drug target selection process is the determination of druggable proteins, and the goal of this selection is identifying drug targets that can be targeted with drug-like chemical compounds. Druggable targets were identified using BLAST algorithm in Drugbank database [51], and the E-value was chosen as $10^{-25}$ [18]. (iii) Broad spectrum analysis was performed for the identification of drug targets that are broadly distributed among other bacteria. Broad spectrum analysis is very beneficial in order to determine drug targets that can be effective against co-infections or multiple infections. In addition, the proteins that are broadly distributed among other bacteria may indicate low mutation rate, so developing antibacterial resistance can be harder for the targeted bacteria. Broad spectrum analysis was performed using PBIT web browser [66], which contains protein sequences of 181 pathogenic organisms. E-value cut-off of $1x10^{-5}$, bit score of 100 and sequence identity of 35% were chosen as parameter values [18, 66]. For the broad spectrum target determination criteria, targets available in at least 40 pathogenic strains were chosen [18].

## Homology modeling of target protein and active / binding pocket prediction

As the 3D structure of target protein was not available in the Protein Data Bank (PDB) [67], the homology modeling was performed to model the target protein's 3D structure. In the first step, the appropriate template protein was searched by BLASTp against the PDB database. The template that matches the criteria of query coverage $\geq$ 90% and percent identity $\geq$ 70% was selected. The Modeller software v9.20 was used to perform comparative protein structure modelling [68–71]. The Modeller works by satisfying the spatial restraints along with employing specific geometrical calculations generating possible coordinate(s) for the location of each single atom of target protein [70]. The homology modeling was performed through Modeller

by implementing a series of python script(s), which resulted in generating five models with their Discrete Optimized Potential Energy (DOPE) values. The 3D model with the lowest DOPE value was selected as the predicted 3D structure of the target protein. The predicted 3D model was then validated through verify3D and Ramachandran Plot via PROCHECK server to ensure its reliability and quality. The Verify3D program was used to interpret the quality of the built model, as Verify3D computes the compatibility of 3D model of target protein against its amino acid sequence [72]. Further, Ramachandran Plot was analyzed to assess the stereo-chemical properties of the predicted 3D model [73].

The built target protein model was subjected to DoGSiteScorer to identify potential binding pockets. The DoGSiteScorer is an automated grid-based program which utilizes difference of Gaussian filter to identify the potential binding pocket present within the protein [74, 75]. The program provides ten potential binding pockets with druggability scores. Out of the ten predicted pockets, the one with the highest druggability score is supposed to be a rich binding pocket and therefore, selected for study. The program also evaluates the depth, surface area and volume for each predicted binding pocket.

## Docking-based virtual screening

To carry out the rigorous virtual screening of drug-like molecules against the target protein, a library of chemical compounds was curated. The ZINC15 database was used to retrieve the drug-like molecules following certain criteria of compound's molecular weight and logP values [76]. The drug-like compounds having logP value ≤ 5 and Molecular Weight ≤ 375 Daltons were obtained from the database in SDF format. A total of 54,000 drug-like compounds were compiled and prepared into the required PDBQT format by using Open Babel. All the 54,000 compounds were minimized with force field MMFF94 via Open Babel [77].

The AutoDock-GPU [78] was chosen to execute molecular docking and virtual screening of curated drug-like molecules (ligands) library against the target protein (P12680). The Auto-Dock GPU was chosen because of the prolonged execution times whilst using AutoDock4. AutoDock-GPU, which is an OpenCL and cuda based implementation of Autodock4, was created to utilize large number of GPU cores and speed up docking by using parallel processing [79, 80]. So, to execute molecular docking, the target protein's modelled 3D structure was prepared by adding hydrogens and its conversion into the required format of PDBQT. After-wards, the grid search box dimensions were set carefully to cover the predicted binding site retrieved from the DoGSiteScorer. Further docking steps were carried out to create docking parameter files, and, eventually, the grid maps FLD file, docking parameters files, GPF and DPF were made. Once the necessary files were created, the molecular docking was performed to screen all the 54,000 compounds with 20 Genetic Algorithm (GA) runs against target receptor's binding site. The virtual screening yielded binding free energy for 20 runs of all the 54,000 compounds. Ultimately, the lowest binding energy of each compound was extracted for structural and functional evaluation. Moreover, re-docking step was performed by execution of all the Autodock4 steps utilizing only one ligand against the target protein to validate predicted binding site.

LigPlot+ program was used to generate 2D diagrams of the target protein and ligand complex. For the LigPlot+ analysis, protein-ligand complex in PDB format was taken. The atomic interaction(s) within the diagram is shown, where ligand and protein's interactive residues are represented in ball-stick format. From the diagram, the amino acid residues of the target protein making chemical interactions with the ligand can be identified. The Lig-Plot+ highlights the hydrogen bonding within the atoms of protein and ligand with green dotted lines.

## Supporting information

**S1 Fig. Metabolic network of pabB related metabolites.** The red arrows indicate the reaction controlled by pabB (L-Glutamine + chorismate -> L-Glutamate + 4-amino-4-deoxychorismate).
(TIF)

**S2 Fig. The predicted 3D structure of P12680.**
(TIF)

**S3 Fig. Ramachandran plot of the predicted 3D structure of target protein (P12680).**
(TIF)

**S4 Fig. The predicted binding pocket is shown as surface in sand brown color while the protein chain as ribbon in blue color.** From the figure, it can be seen that $Mg^{2+}$ ion is submerged within the predicted binding pocket.
(TIF)

**S5 Fig.** A) is representing the LigPlot+ of P12680 showing atomic interaction between protein (residues/ $Mg^{2+}$ ion) and ligand (ZINC7879733). The atomic linkages due to hydrogen bonding can be identified from the diagram. Similarly, B) represents the LigPlot+ analysis of P12680 and ligand (ZINC15179659), C) represents the LigPlot+ for P12680 and ligand (ZINC14880941), and D) represents the LigPlot+ for P12680 and ligand (ZINC58542694).
(TIF)

**S6 Fig.** A) is the LigPlot+ for P12680 and ligand (ZINC1201089024), all the atomic linkages occurring between the protein-ligand complex can be analyzed. Likewise, B) represents the LigPlot+ for P12660 and ligand (ZINC27071723), C) is LigPlot+ for P12680 and ligand (ZINC7133393) and D) is LigPlot+ for P12680 and ligand (ZINC7879735).
(TIF)

**S7 Fig.** A) is the LigPlot+ for P12680 and ligand (ZINC58542238) complex and B) is the LigPlot+ for P12680 and ligand (ZINC7538530) complex. The overall amino acid residues of P12680 which interact with each of the top ten compounds (ligands) making hydrogen bonds and hydrophobic contacts.
(TIF)

**S8 Fig. Principal component analysis (PCA) of GSE117236.** The red, blue and green dots represent beginning of infection, 8[th] hour of infection and 16[th] hour of infection respectively.
(TIF)

**S9 Fig. Box plots of gene expression values of HeLa cell and *S*. Typhimurium.**
(TIFF)

**S1 Table. 140 essential genes based on DEG database.**
(DOCX)

**S2 Table. 89 potential drug targets that are not similar to human proteins.**
(DOCX)

**S3 Table. Potential drug targets that have high affinity to bind drug-like molecules.**
(DOCX)

**S4 Table. The list of final potential drug targets.** The last column indicates broad spectrum analysis results.
(DOCX)

**S5 Table. Binding energy, Zinc IDs, 1D and 2D structure of each of top 10 compounds.**
(DOCX)

**S6 Table. Metabolites that can be consumed by *S.* Typhimurium inside host cytoplasm in pathogen-host GMN model.**
(DOCX)

**S7 Table. The exchange metabolites of *S.* Typhimurium that match with the cytoplasmic metabolites of human cell.**
(DOCX)

**S8 Table. DMEM medium constraints for the host GMN used in metabolic model simulations.**
(DOCX)

**S9 Table. Long names of metabolites given in S1 Fig, the figüre that reports metabolite-metabolite interactions around pabB catalyzed reaction.**
(DOCX)

**S1 File. The MATLAB codes to generate and analyze pathogen-host integrated genome scale metabolic models of *S.* Typhimurium and human.**
(ZIP)

## Author Contributions

**Conceptualization:** Reaz Uddin, Tunahan Çakır.

**Data curation:** Kadir Kocabaş.

**Formal analysis:** Kadir Kocabaş, Alina Arif, Reaz Uddin, Tunahan Çakır.

**Methodology:** Kadir Kocabaş, Alina Arif, Reaz Uddin.

**Project administration:** Tunahan Çakır.

**Supervision:** Tunahan Çakır.

**Writing – original draft:** Kadir Kocabaş, Alina Arif.

**Writing – review & editing:** Reaz Uddin, Tunahan Çakır.

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
