## [Decision Letter · Decision Letter 0]

17 Feb 2022

PONE-D-21-39848Dual transcriptome based reconstruction of Salmonella-human integrated metabolic network to screen potential drug targetsPLOS ONE

Dear Dr. Cakir,

Thank you for submitting your manuscript to PLOS ONE. After careful consideration, we feel that it has merit but does not fully meet PLOS ONE’s publication criteria as it currently stands. Therefore, we invite you to submit a revised version of the manuscript that addresses the points raised during the review process.

While one reviewer was convinced by your current manuscript, the other expert reviewer raised some critical points which should be adequately addressed in a revised manuscript.

We look forward to receiving your revised manuscript.

Kind regards,

Roman G. Gerlach

Academic Editor

PLOS ONE

Journal Requirements:

“This work was supported by TUBITAK, The Scientific and Technological Research Council of Turkey (Project Code: 316S005) and by PSF, The Pakistan Science Foundation [Project Code: PSF-TUBITAK/S-HEJ (04)].”

“This work was supported by TUBITAK, The Scientific and Technological Research Council of Turkey (Project Code: 316S005) and by PSF, The Pakistan Science Foundation [Project Code: PSF-TUBITAK/S-HEJ (04)]. The funders had no role in study design, data collection and analysis, decision to publish, or preparation of the manuscript.”

Reviewers' comments:

Reviewer's Responses to Questions

**Comments to the Author**

1. Is the manuscript technically sound, and do the data support the conclusions?

Reviewer #1: Yes

Reviewer #2: Partly

2. Has the statistical analysis been performed appropriately and rigorously? 

Reviewer #1: I Don't Know

Reviewer #2: I Don't Know

3. Have the authors made all data underlying the findings in their manuscript fully available?

Reviewer #1: Yes

Reviewer #2: Yes

4. Is the manuscript presented in an intelligible fashion and written in standard English?

Reviewer #1: Yes

Reviewer #2: Yes

5. Review Comments to the Author

Reviewer #1: In this manuscript, Kocabas et al took advantage of the available dual RNA-seq data from Salmonella-infected cells, and constructed in silico an integrated metabolic network for Salmonella and host cells. Based on the metabolic network, the authors predicted 140 essential bacterial metabolic genes and further identified pabB as a potential druggable target. In silico docking analysis suggested PabB might interact with small molecules for drug development.

Though most of the analysis were purely in silico with little credibility, this is an innovative approach and an interesting idea in my personal opinion. The study provides additional angles to decode other RNA-seq data and may help promote the understanding of Salmonella infection processes.

Reviewer #2: Kocabaş et al report the reconstruction of transcriptome-specific pathogen-host integrated models using dual RNA-seq data of Salmonella Typhimurium and infected HeLa cells, which predicts pabB gene of the pathogen as one of the top druggable targets. The authors have also performed structure-based virtual screening of 54,000 compounds against pabB to identify 1659 compounds, leading to a final top 10 best-ranking compounds. The manuscript highlights the use of dual RNA seq based metabolic modeling as an approach to identify drug targets - which has scientific merit - however it would benefit by considering some experimental validations and additional performance assessment of the model predictions. The host (HeLa cell) metabolic model, iHsa (Blais et al, 2017) and the pathogen model, stm_v1.0 (Thiele et al 2011) are already published ones and the concept of pathogen-host integration is also reported already for Mycobacterium (Bordbar et al 2010, Rienksma et al 2019). The novelty of this work is the use of dual RNA seq to constrain the metabolic model, that is based on GIMME algorithm (Becker et al 2008), which is well-known for developing cell-specific models. Considering the overlap between reported concepts, it requires rigorous performance assessments to make it condition-specific and to be considered as the first application to Salmonella.

Comments:

• The analyses of the model performance could have been more extensive while comparing with experimental validations (For e.g., only secretion of 3 metabolites are compared in the manuscript – more such predictions need to be validated - in the host-side of the network as well).

• The model assessment for precision and recall, to deduce its accuracy of gene essentiality prediction needs to be performed in comparison with the experimental datasets (e.g., mutants in Salmonella for the identified gene sets?) or literature evidence to categorize the predictions as “true positives” and “false positives”. Among the 140 gene targets from gene essentiality predictions, how many of them have literature evidence as essential for Salmonella inside host (HeLa) cells or are all novel predicitons?

• The prioritization criteria for selecting pabB from 28 potential drug targets - that are non-homologous to human proteins, druggable and broadly distributed among other bacteria, is only based on the importance for PABA. This should be justified by mentioning how they arrived at one candidate gene based on the model predictions, for e.g., effect in biomass within host (% inhibition) or participation in higher number of essential reactions in the network?

• Metabolic modeling could be used to track the mechanism of a knockout phenotype. Could the authors sketch out the mechanism by which pabB is becoming essential in Salmonella? via metabolic network – delineating the reactions involving pabB and the effect of deleting pabB in Salmonella? How many metabolic reactions in pathogen network are linked to pabB and/or other gene targets identified? How many of these reactions become essential reactions for the pathogen within host?

• Line 383, Equation 1 - α and β are calculated via FBA and then used as weights. Will it be limited to the constraints used each time the FBA is run or is fixed for all simulations? Instead, could FBA with two objective functions (HB and PB) at the same time, by changing objective function weights would be more appropriate?

• Top 10 compounds identified and listed in Table 4, requires more explanation on the status of their current applications – any homology with natural products, or in clinical trial or used in other bacterial screening etc., in order to increase the application of these identified compounds.

• Experimental screening for at least 3 compounds is required to support the claim that it can be a useful drug against Salmonella inside host cell. For e.g., Treating the Hela cells infected with Salmonella in vitro should inhibit the growth of the pathogen as predicted. Or finding the binding specificity of these compounds for pabB gene.

• Figure 1 would benefit by having two panels – separately for A) host and B) pathogen and having the column bar side by side to represent 0th and 16th hour instead of superimposing them.

• The dataset used also has an 8h time point. Why was it excluded from the analysis?

6. PLOS authors have the option to publish the peer review history of their article (what does this mean?). If published, this will include your full peer review and any attached files.

Reviewer #1: No

Reviewer #2: No

---

## [Author Response · Author response to Decision Letter 0]

4 Apr 2022

We have updated the manuscript based on the editorial comments by removing Acknowledgement section and by making it compatible with PLOS One formatting styles. We include our responses to reviewer comments below.

Replies for Reviewer Comments

Reviewer 2:

1) The analyses of the model performance could have been more extensive while comparing with experimental validations (For e.g., only secretion of 3 metabolites are compared in the manuscript – more such predictions need to be validated - in the host-side of the network as well).

We agree with the reviewer that additional tests on the model performance would strengthen the manuscript. On the other hand, the studies that report measured reaction fluxes of host and pathogen during infection of S. Typhimurium are very limited in the literature. We could only find information about the secretion rates of these three pathogen metabolites to evaluate the performance of the model. To provide extra validation of the model -as recommended by the reviewer in the next comment- we additionally compared our gene essentiality predictions with the literature (details given in the next reply). We believe the metabolite secretion prediction and gene essentiality prediction performances provide evidences on the suitability of our pathogen-host integrated genome-scale metabolic network.

2) The model assessment for precision and recall, to deduce its accuracy of gene essentiality prediction needs to be performed in comparison with the experimental datasets (e.g., mutants in Salmonella for the identified gene sets?) or literature evidence to categorize the predictions as “true positives” and “false positives”. Among the 140 gene targets from gene essentiality predictions, how many of them have literature evidence as essential for Salmonella inside host (HeLa) cells or are all novel predictions?

We are grateful to the reviewer for this comment, which enabled us to provide an additional source of validation for our pathogen-host integrated genome-scale metabolic model. Predicted essential genes were checked by using Database of Essential Genes (DEG), which reports data from three experimental gene deletion studies from rich medium experiments (Barquist et al., 2013; Khatiwara et al., 2012; Knuth et al., 2004). The results were provided in the revised manuscript under the section “Identification of Potential Drug Targets”, first paragraph and 3rd paragraph. Data were available for 137 of 140 predicted essential genes, 93 of which were reported as essential genes in at least one study (68%). Regarding the prioritized list of 28 essential genes, 20 were reported as essential in the database. When we ranked the prioritized list in terms of broad spectrum score (number of pathogenic bacteria with significantly similar sequence of the gene), 16 of the top 20 genes were essential based on the DEG database (80%). Considering that the experiments were performed in rich medium conditions, with no host cells involved, we believe the predictions provide additional validation of the pathogen-host integrated genome-scale metabolic network. We have also added a new column to Table 3, indicating whether the predicted drug target is reported to be essential in DEG.

3) The prioritization criteria for selecting pabB from 28 potential drug targets - that are non-homologous to human proteins, druggable and broadly distributed among other bacteria, is only based on the importance for PABA. This should be justified by mentioning how they arrived at one candidate gene based on the model predictions, for e.g., effect in biomass within host (% inhibition) or participation in higher number of essential reactions in the network?

All of the proposed candidate drug targets were selected based on their effect on biomass production. Inhibition of all the proposed 28 candidate drug targets leads to blocking of biomass production (zero growth rate). pabB is also reported as an essential gene in DEG database. 20 of 28 candidate targets were experimentally verified in the DEG database. When we rank those 20 genes in terms of broad spectrum behaviour, pabB is 7th in terms of number of pathogenic bacteria strains that carry a gene with high sequence similarity. The reaction associated with pabB is related to production of nucleic acids thymine and adenine as well as production of numerous amino acids in the model. Therefore, we chose pabB for docking analysis. The network that shows the relationship of pabB with critical metabolites was given as Supplementary Figure S1 in the revised manuscript (please also see our reply to the next comment). Those explanations are now included in the revised manuscript, Section “Identification of potential drugs for pabB”, as given below. We thank the reviewer for helping us clearing up the ambiguity associated with this issue in our manuscript. 

“Additionally, we investigated the importance of pabB in the pathogen-host integrated GMN model. Interactions of the metabolites of the reaction (chorismate, 4-amino-4-deoxychorismate, L-glutamate, L-glutamine) with the metabolites of other reactions were visualized by creating a metabolite-metabolite interaction network (S1 Fig.). L-glutamate is directly related to numerous amino acids such as alanine, leucine and asparagine. On the other hand, 4-amino-4-deoxychorismate is indirectly related with the production of thymidine through tetrahydrofolate. It is also linked with the production of adenine through R-Pantoate. Therefore, DNA synthesis is dependent on the production of 4-amino-4-deoxychorismate through adenine and thymine synthesis, which cannot be synthesized when pabB is inhibited.”

4) Metabolic modeling could be used to track the mechanism of a knockout phenotype. Could the authors sketch out the mechanism by which pabB is becoming essential in Salmonella? via metabolic network – delineating the reactions involving pabB and the effect of deleting pabB in Salmonella? How many metabolic reactions in pathogen network are linked to pabB and/or other gene targets identified? How many of these reactions become essential reactions for the pathogen within host?

We thank again the reviewer for this comment. We investigated the effect of inhibition of pabB through metabolite-metabolite interaction network, which is indeed a feature implemented in COBRA Toolbox by one of the authors (K. Kocabas). The network is given as a supplementary figure for the revised manuscript. The results and comments were provided in the revised manuscript under the section “Identification of potential drugs for pabB”. Briefly, we tracked the metabolites that were affected from to function of pabB. Since the associated reaction is related to several amino acids as well as two nucleic acids, its blocking inhibits macromolecular synthesis, leading to zero growth.

5) Line 383, Equation 1 - α and β are calculated via FBA and then used as weights. Will it be limited to the constraints used each time the FBA is run or is fixed for all simulations? Instead, could FBA with two objective functions (HB and PB) at the same time, by changing objective function weights would be more appropriate?

We used fixed α and β in all simulations. We deliberately did not change the objective function since then it would be ambiguous to identify the source of change..

6) Top 10 compounds identified and listed in Table 4, requires more explanation on the status of their current applications – any homology with natural products, or in clinical trial or used in other bacterial screening etc., in order to increase the application of these identified compounds.

There are not any literature reports for the biological activities of the 10 identified compounds. However, the shortlisted compounds are drug-like as they obey the Lipinski rules. In addition, these compounds were derived from drug-like library i.e. the ZINC15 database. Moreover, to focus on only drug-like compounds, we further filtered the compounds in the database and only obtained the compounds having logP value ≤ 5 and Molecular Weight ≤ 375 daltons from this database, which is stated in Materials & Methods section.

7) Experimental screening for at least 3 compounds is required to support the claim that it can be a useful drug against Salmonella inside host cell. For e.g., Treating the Hela cells infected with Salmonella in vitro should inhibit the growth of the pathogen as predicted. Or finding the binding specificity of these compounds for pabB gene.

We agree that experimental validation would strengthen our theoretical findings of 10 candidate compounds. On the other hand, our research groups do not have any experimental lab facility, we are purely computational labs. We believe our major contribution to the scientific field with this work is (i) to provide a validated pathogen-host genome-scale metabolic network for S. Typhimurium, (ii) to demonstrate the integrative analysis of dual RNA-Seq data with pathogen-host genome-scale metabolic models. We have included in the revised manuscript a statement that the compounds need experimental validation (Last sentences of section “Identification of potential drugs for pabB”.

On the other hand, the following paper from our group has been well cited as well as found experimental proof within one month of this article published online:

Ref#1: Identification of chymotrypsin-like protease inhibitors of SARS-CoV-2 via integrated computational approach, Journal of Biomolecular Structure and Dynamics, (2021), 39:7, 2607-2616, DOI: 10.1080/07391102.2020.1751298

We believe our theoretical findings will still help other researchers in drug development field, and will aid in the efforts to design novel drugs. 

8) Figure 1 would benefit by having two panels – separately for A) host and B) pathogen and having the column bar side by side to represent 0th and 16th hour instead of superimposing them.

We thank again the reviewer for this comment. We recreated the Figure 1 as panel A) and B). We used bidirectional bar chart instead of creating a bar chart with column bar side by side to represent 0th and 16th since it otherwise led to a too big figure that contained a lot of empty spaces. 

9) The dataset used also has an 8h time point. Why was it excluded from the analysis?

We wanted to compare the beginning of the infection with the late stage of the infection. We could not see any considerable differences between 0 time point and 8h time point in terms of predicted flux values. Probably the infection does not yet show its damage to the cells in this time point. Therefore, we did not include 8h time point. We added this information in the revised manuscript under the Materials&Methods section “Transcriptome data”.

---

## [Decision Letter · Decision Letter 1]

4 May 2022

PONE-D-21-39848R1Dual transcriptome based reconstruction of Salmonella-human integrated metabolic network to screen potential drug targetsPLOS ONE

Dear Dr. Cakir,

Thank you for submitting your manuscript to PLOS ONE. After careful consideration, we feel that it has merit but does not fully meet PLOS ONE’s publication criteria as it currently stands. Therefore, we invite you to submit a revised version of the manuscript that addresses the points raised during the review process.

I apologize for the delayed review process.

I agree with the 2^nd^ reviewer that it is worth to include the transcriptome data of the 8h time point in the manuscript. With that included, your manuscript can be accepted for publication.

We look forward to receiving your revised manuscript.

Kind regards,

Roman G. Gerlach

Academic Editor

PLOS ONE

Journal Requirements:

Reviewers' comments:

Reviewer's Responses to Questions

**Comments to the Author**

1. If the authors have adequately addressed your comments raised in a previous round of review and you feel that this manuscript is now acceptable for publication, you may indicate that here to bypass the “Comments to the Author” section, enter your conflict of interest statement in the “Confidential to Editor” section, and submit your "Accept" recommendation.

Reviewer #1: All comments have been addressed

Reviewer #2: (No Response)

2. Is the manuscript technically sound, and do the data support the conclusions?

Reviewer #1: Yes

Reviewer #2: Yes

3. Has the statistical analysis been performed appropriately and rigorously? 

Reviewer #1: Yes

Reviewer #2: Yes

4. Have the authors made all data underlying the findings in their manuscript fully available?

Reviewer #1: Yes

Reviewer #2: Yes

5. Is the manuscript presented in an intelligible fashion and written in standard English?

Reviewer #1: Yes

Reviewer #2: Yes

6. Review Comments to the Author

Reviewer #2: The authors have addressed the comments. However, it will be worthwhile to show the data mentioned in Comment -9. It is important to see the early changes (8h time point) in the metabolic flux irrespective of how small they are.

---

## [Author Response · Author response to Decision Letter 1]

5 May 2022

The number of reaction and metabolite information of the reconstructed GMN model for the early infection (8h time point) were added to Table 1 in the Results section. The predicted by-product secretion rates based on this metabolic model were also added to Table 2. We also updated Supplementary Figure 8 (PCA of dual transcriptome data) by including the samples of 8th hour of infection.

---

## [Editor Report · Decision Letter 2]

11 May 2022

Dual transcriptome based reconstruction of Salmonella-human integrated metabolic network to screen potential drug targets

PONE-D-21-39848R2

Dear Dr. Cakir,

We’re pleased to inform you that your manuscript has been judged scientifically suitable for publication and will be formally accepted for publication once it meets all outstanding technical requirements.

Kind regards,

Roman G. Gerlach

Academic Editor

PLOS ONE
---

## [Editor Report · Acceptance letter]

16 May 2022

PONE-D-21-39848R2 

Dual transcriptome based reconstruction of Salmonella-human integrated metabolic network to screen potential drug targets 

Dear Dr. Cakir:

I'm pleased to inform you that your manuscript has been deemed suitable for publication in PLOS ONE. Congratulations! Your manuscript is now with our production department. 

Kind regards, 

on behalf of

Dr. Roman G. Gerlach 

Academic Editor

PLOS ONE